# Integrated Analysis of Metabolome and Transcriptome Revealed Different Regulatory Networks of Metabolic Flux in Tea Plants [*Camellia sinensis* (L.) O. Kuntze] with Varied Leaf Colors

**DOI:** 10.3390/ijms25010242

**Published:** 2023-12-23

**Authors:** Yazhen Zhang, Liyuan Wang, Xiangrui Kong, Zhihui Chen, Sitong Zhong, Xinlei Li, Ruiyang Shan, Xiaomei You, Kang Wei, Changsong Chen

**Affiliations:** 1Tea Research Institute, Fujian Academy of Agricultural Sciences, Fuzhou 350012, China; zhangyazhen_ada@163.com (Y.Z.); kongxiangrui_2008@163.com (X.K.); chenzhihui75@sina.com (Z.C.); zhongsitong1110@163.com (S.Z.); lxlfafu@163.com (X.L.); fjnkycys@163.com (R.S.); yxm0593@163.com (X.Y.); 2Key Laboratory of Tea Biology and Resources Utilization, Ministry of Agriculture, National Center for Tea Improvement, Tea Research Institute Chinese Academy of Agricultural Sciences (TRICAAS), Hangzhou 310008, China; wangly@tricaas.com

**Keywords:** *Camellia sinensis*, leaf color, metabolome, transcriptome, lipid, flavonoid

## Abstract

Leaf color variations in tea plants were widely considered due to their attractive phenotypes and characteristic flavors. The molecular mechanism of color formation was extensively investigated. But few studies focused on the transformation process of leaf color change. In this study, four strains of ‘Baijiguan’ F1 half-sib generation with similar genetic backgrounds but different colors were used as materials, including Green (G), Yellow-Green (Y-G), Yellow (Y), and Yellow-Red (Y-R). The results of broadly targeted metabolomics showed that 47 metabolites were differentially accumulated in etiolated leaves (Y-G, Y, and Y-R) as compared with G. Among them, lipids were the main downregulated primary metabolites in etiolated leaves, which were closely linked with the thylakoid membrane and chloroplast structure. Flavones and flavonols were the dominant upregulated secondary metabolites in etiolated leaves, which might be a repair strategy for reducing the negative effects of dysfunctional chloroplasts. Further integrated analysis with the transcriptome indicated different variation mechanisms of leaf phenotype in Y-G, Y, and Y-R. The leaf color formation of Y-G and Y was largely determined by the increased content of eriodictyol-7-O-neohesperidoside and the enhanced activities of its modification process, while the color formation of Y-R depended on the increased contents of apigenin derivates and the vigorous processes of their transportation and transcription factor regulation. The key candidate genes, including *UDPG*, *HCT*, *CsGSTF1*, *AN1/CsMYB75*, and *bHLH62*, might play important roles in the flavonoid pathway.

## 1. Introduction

As an important economic plant for leaf use, tea is one of the most widely consumed beverages worldwide. Generally, tea is a woody plant with perennial evergreen leaves. However, in order to adapt to different growth environments and the diverse demands of consumers, it has undergone plentiful mutations during the long history of natural evolution and artificial selection [1]. Due to the visible characteristic, leaf phenotypes with different mutated colors were widely considered by scholars. Etiolated (yellow), albino (white), and purple or red leaves were the common variations in tea plants and acted as high-quality breeding materials [2]. The mutation mechanisms were also widely investigated due to their attractive phenotypes and health benefits. Numerous studies have shown that the contents and compositions of chlorophyll, flavonoids, and carotenoids directly affect the color formation of tea plants [3]. Chlorophylls are essential for green color formation. Flavonoids include anthocyanins, catechins, flavones, and flavonols, which are important pigments for red, pink, and purple-blue phenotypes in plants. Carotenoids are orange, yellow, and red-related pigments [4]. It has been reported that the yellow to albino tea leaves have defective chloroplasts, lower chlorophyll and carotenoid contents, and suppress flavonoid biosynthesis [5,6]. Quality-related compounds in tea leaves also change coordinately. Generally, yellow or albino tea leaves accumulate more free amino acids (especially theanine) and fewer catechins [7,8]. While the red or purple phenotype of tea leaves was attributed to increased anthocyanins and decreased chlorophylls [4,9]. As mentioned above, numerous investigations were performed on tea plants with different leaf colors. The mechanisms of leaf variation from green to yellow (albino) or red (purple) were basically understood. Unfortunately, the variation mechanism of leaf color was usually studied in tea plants with different genetic backgrounds, which largely interfered with the results. 

‘Baijiguan’ is a light-sensitive albino tea cultivar from Wuyi mountain in Fujian Province and has a history of more than 300 years. It is an important parent for genetic breeding and a key germplasm for revealing the regulatory mechanisms of flavor and quality-related compounds [10]. The tender shoots of ‘Baijiguan’ exhibited a yellow phenotype under normal light conditions, which was caused by reduced chlorophyll and carotenoid content. The leaf color could convert from yellow to green under shade treatment [11]. Interestingly, the following study revealed that the transformation of leaf color in ‘Baijiguan’ was regulated by light intensity through altering the contents and proportions of lipids, which provided new insights into the variation mechanism of tea leaves [12]. In addition, different from other albino tea cultivars with increased free amino acid levels and decreased catechin levels, there are no significant differences in albino half-sibs of ‘Baijiguan’ [13]. It indicated that there might be a special metabolic network in ‘Baijiguan’ and its offspring. The comprehensive regulatory mechanism needs to be further explored. 

In this study, F1 half-sib offspring were generated through open pollination of ‘Baijiguan’, which acted as the maternal parent. Four strains under identical natural environmental conditions with similar genetic backgrounds but different leaf colors were used as materials. An integrated analysis of the widely targeted metabolome and transcriptome was applied. Key metabolic pathways and genes were determined to explain the varied phenotypes of tea leaves. This research will enhance our understanding of the metabolic networks and molecular mechanisms of leaf color variations in tea plants.

## 2. Results

### 2.1. The Phenotype and Relative Chlorophyll Concentration of Tea Leaves Varied Coordinately

Phenotype variances in leaf colors were observed in the four different strains of ‘Baijiguan’ offspring. As shown in Figure 1a, the leaf colors from left to right were green (G), yellow-green (Y-G), yellow (Y), and yellow-red (Y-R). The relative chlorophyll concentration was measured by soil and plant analyzer development (SPAD) and displayed as a numerical value. The SPAD value of the leaf samples ranged from 2.51 to 17.93 (Figure 1b). The highest SPAD value was found in G, which was 3.2–7.1 folds higher than those in other strains. The SPAD value in Y-R was significantly higher than that in Y-G and Y. While no significant SPAD value difference was observed between Y-G and Y, the results of the SPAD value were consistent with the phenotype of tea leaves.

### 2.2. The Cellular Ultrastructure of Tea Leaves with Different Colors

To further compare the cellular ultrastructure in tea leaves with different colors, a transmission electron microscope (TEM) analysis was conducted (Figure 2). The results showed that the structure of the cell and chloroplast (green circle and arrow in Figure 2) in G was normal and complete. There were typical cellular characteristics with a central vacuole (red arrows in Figure 2a–d), a well-established chloroplast, and a clear thylakoid (Figure 2e–h) in G. However, all of the yellow leaves (Y-G, Y, and Y-R) exhibited abnormal morphology in both the cell and chloroplast. Smaller vacuoles and more vesicles (yellow arrows in Figure 2a–d) were observed. The chloroplast showed lower abundance, looser-arranged grana lamella, and defective thylakoid (Figure 2e–h). Interestingly, compared with Y-G and Y, the number of vesicles in Y-R has increased anomalously (Figure 2d). The structure of grana lamella was also recovered partially (Figure 2h). In brief, compared with G, the cell membrane and compartmentalization in yellow leaves were damaged (Y-R) or disrupted (Y-G and Y), which are consistent with their phenotype and SPAD value.

### 2.3. Differentially Accumulated Metabolites (DAMs) in Green and Yellow Leaves

To investigate the metabolite changes in varied-color tea leaves, broadly targeted metabolome technology based on the UPLC-MS/MS system was used in our study. Totally 1146 metabolites were detected in the tested samples. They could be divided into 11 classes, including amino acids and derivatives (102), nucleotides and derivatives (65), organic acids (85), lipids (114), phenolic acids (213), flavonoids (299), tannins (27), alkaloids (85), lignans and coumarins (43), terpenoids (10), and others (103). The levels of these metabolites are displayed as a heatmap in Figure 3a. Different accumulated patterns of metabolites were found in varied-color leaves. Groups of Y-G and Y showed similar trends, while groups of G and Y-R showed greater differences. The results of PCA revealed that three biological replicates of each group were clustered closely. The four group samples could be distinguished clearly from each other. Moreover, Y-G and Y were also found to be closely together, which suggests their similarities in metabolic profiling (Figure 3b). 

Further DAM analysis was conducted using the G group as control. In total, 185, 197, and 206 DAMs were identified by comparing with Y-G, Y, and Y-R, respectively. For G vs. Y-G and G vs. Y, the number of downregulated DAMs (130 and 109) was higher than that of the upregulated DAMs (55 and 88). Most DAMs were detected in G vs. Y-R, with 104 downregulated and 102 upregulated metabolites (Figure 3c). All of the DAMs were then subjected to Venn analysis. As shown in Figure 3d, a total of 389 DAMs were detected, and 47 common DAMs were found to be accumulated differentially in all three comparison groups. 

The detailed distribution patterns of the 47 DAMs are displayed in Figure 4. More than half of the 47 DAMs were classified into primary metabolites (lipids) and secondary metabolites (flavonoids). The 10/11 lipids were most enriched in G, including 4 free fatty acids, 2 glycerol esters, and 4 lysophosphatidyl ethanolamine (LysoPE). Significant decreases with 3.85–5.60 fold changes of 4 LysoPE were found in yellow leaves. However, 12/13 flavonoids showed higher contents in the yellow leaves (Y-G, Y, and Y-R), including 8 flavones and 4 flavonols. Most of the flavones, such as apigenin-7-O-glucoside (Lmpp003930), apigenin-7-O-rutinoside (pme0368), luteolin-7-O-rutinoside-5-O-rhamoside (Hmmn003584), tricin (Lmzp004885), and jaceosidin (pmp000004), showed higher levels in Y-R. While most of the flavonols, such as kaemferol-3-O-galactoside (Lmyp004318), kaempferol-3-O-arabinoside-7-O-rhamnoside (Hmcp001858), and isohamnetin (mws0066), showed higher contents in Y. Interestingly, the content of eriodictyol-7-neohesperidoside (Hmmn004152) belonging to flavones in Y was 2.9–14.3 folds higher than that in other groups. The content of 7-methyl-kaempferol (Lmjp004941), a type of flavonol, showed the highest level in Y-R. These results indicate that the flux of primary and secondary metabolism changed significantly in green and yellow leaves.

### 2.4. Differentially Expressed Genes (DEGs) in Green and Yellow Leaves

RNA-seq analysis was performed to uncover the molecular mechanism of leaf color formation in tea plants. Twelve cDNA libraries were separately established using the four groups of tea samples. In total, 596.88 million raw reads and 537.79 clean reads were obtained. Each sample generated 6.02–7.8 G of clean base with Q20 and Q30 over 97% and 92%, respectively. The GC content ranged from 43.07 to 43.86% (Appendix A). Then the clean reads were mapped to the ‘Tieguanyin’ tea genome. The ratio of mapped reads ranged from 86.97 to 88.55%. Among them, 82.54–84.22% of the reads were uniquely mapped to the genome (Appendix A). 

In order to compare the gene expressions among different samples, the gene expression level was determined by fragments per kilobase of transcript per million fragments mapped (FPKM). The PCA analysis of detected gene expression levels revealed that the three replicates of G and Y-R clustered separately, while groups Y-G and Y showed similar patterns and failed to distinguish from each other (Figure 5a). Further different expression gene (DEG) analysis was performed using the G group as control. A total of 11,793 DEGs were obtained from the three comparisons. The expression levels of DEGs were displayed as a heatmap in Figure 5b. The result was consistent with the PCA analysis. Y-G and Y were clustered as a subgroup, and the Y-R group was clearly separated from the other groups. The number of DEGs in each comparison is shown in Figure 5c. The most DEGs were identified in G vs. Y-R, with 2787 downregulated and 3381 upregulated genes, while the comparison G vs. Y-G showed the fewest DEGs, with 1998 downregulated and 2069 upregulated genes. All of the DEGs were then subjected to Venn analysis. Totally 1004 co-regulated genes were found to be expressed differentially in the three comparisons (Figure 5d).

In addition, to confirm the reliability of RNA sequencing, quantitative real-time PCR (qRT-PCR) analysis was conducted on 15 candidate DEGs involved in leaf-color-related pathways, including photosynthesis and the metabolisms of chlorophyll, carotenoid, and flavonoid. As shown in Figure 6, the selected gene expression patterns in qRT-PCR analysis and transcriptome data were highly consistent. These results indicated that the RNA-seq profiling was reliable and could be used for further analysis. 

### 2.5. DAMs and DEGs Were Enriched in Consistent Metabolic Pathways

According to the results of DAMs and DEGs, a conjoint analysis of the metabolome and transcriptome was then performed to reveal their relationship and regulatory mechanism. Considering the similarity of groups Y-G and Y, one of group Y was selected for further analysis. The top 25 co-enriched KEGG pathways of DAMs and DEGs are displayed in Figure 7. The results showed that glycerolipid metabolism (ko00561), glycerophospholipid metabolism (ko00564), and flavonoid biosynthesis (ko00941) were significantly enriched pathways in both comparisons of G vs. Y and G vs. Y-R (Figure 7a,b). Moreover, the flavonoid biosynthesis pathway was also found to be co-enriched in the comparison of Y vs. Y-R (Figure 7c). It indicated that metabolic flux was changed coordinately along with the leaf color variations. DAMs and DEGs involved in the flavonoid pathway play potentially important roles in the color formation of tea leaves.

### 2.6. Flavonoid Pathway Was Regulated Differentially in Green and Yellow Leaves

Firstly, functional genes involved in flavonoid biosynthesis were screened out, including phenylalanine ammonia-lyase (PAL), cinnamate-4-hydroxylase (C4H), 4-coumaroy CoA ligase (4CL), chalcone synthase (CHS), chalcone isomerase (CHI), flavonoid 3′ hydroxylase (F3′H), flavonoid 3′,5′-hydroxylase (F3′5′H), flavanone-3-hydroxylase (F3H), flavone synthase II (FNS), flavonol synthase (FLS), dihydroflavonol 4-reductase (DFR), anthocyanidin synthase (ANS), and leucoanthocyanidin reductase (LAR) [14]. Candidate structural genes with a maximum FPKM ≥ 50 in the twelve samples were obtained, including PAL (3), C4H (1), 4CL (2), CHS (4), CHI (3), F3H (1), FNS (1), FLS (2), F3′H (1), F3′5′H (1), DFR (2), ANS (1), and LAR (2). As shown in Figure 8a, expression levels of genes upstream of flavonoid biosynthesis decreased in the Y and Y-R, which was inconsistent with the tendency of flavonoid contents. The downstream of the metabolic pathway after naringenin formation also exhibited similar gene expression patterns, except for FNS, F3′H, and F3′5′H. Further correlation analysis between gene expression and flavonoid content was then performed. As shown in Figure 8b and Appendix A, most of the tested genes were negatively correlated with the DAMs of flavones and flavonols. Interestingly, the expressions of FNS and F3′5′H were positively but not significantly correlated with most of the flavones and flavonols. These key genes involved in flavonoid biosynthesis could not completely contribute to the increased accumulation of flavonoids.

Therefore, flavonoid glycosylation and acylation-related genes that were annotated as flavonoid-related pathways (ko00940, ko00941, ko00942, ko00943, and ko00944) were screened out, including *UDP-glycosyltransferase* (*UDPG*) and *hydroxy-cinnamoyl transferase* (*HCT*). Totally 9 *UDPG* and 24 *HCT* genes with a maximum FPKM ≥ 5 in the tested samples were obtained. Their expression patterns were presented as a heatmap in Figure 9a. Overall, higher gene expression levels were found in yellow leaves (Y-G, Y, and Y-R) than those in G. Relationships with 12 DAMs of flavones and flavonols were then investigated. Based on the threshold of correlation coefficient > 0.8 and *p*-value < 0.05, a total of 32 genes related to 11 metabolites were subjected to the construction of a correlation network (Figure 9b). More than half of the genes showed similar relationships with the three flavonoid components. Namely, eriodictyol-7-neohesperidoside (Hmmn004152) was found to be significantly positively correlated with most genes. While two apigenin derivates (Lmpp003930 and pme0368) were found to be significantly negatively correlated with most genes. In total, 12 upregulated DEGs in Y were screened out. Among them, 2 *UDPG* (*TGY105792*, *TGY070785*) and 1 *HCT* (*TGY018791*) genes with the maximum FPKM ≥ 30 in the tested samples correlated with the three components (marked in red font in Figure 9b). These results showed that the genes related to flavonoid glycosylation and acylation were upregulated in yellow leaves, especially in Y-G and Y. It suggested that the more vigorous modification process of flavonoids might be one of the reasons for the formation of yellow leaves in tea plants.

The transport of flavonoids was another important process that affected their final accumulation in plants [15]. Hence, DEGs of the transporter families, including *ATP-binding cassette* (*ABC*), *multidrug and toxin extrusion* (*MATE*), and *glutathione S-transferase* (*GST*), were analyzed in this study. The expression levels of 14 *ABC*s, 6 *MATE*s, and 9 *GST*s with the maximum FPKM ≥ 5 were displayed in Figure 10a. Most of the genes are expressed highly in Y-R. Further correlation analysis showed that a total of 28 genes were significantly correlated with 13 DAMs of flavonoids (Figure 10b). Interestingly, 20 genes were also found to be closely related to the three flavonoid components (Hmmn004152, Lmpp003930, and pme0368), but showed contrary relationships compared with the results of *UPDG* and *HCT*. Among them, 10 DEGs showed consistent relationships, namely negatively correlated with eriodictyol-7-neohesperidoside (Hmmn004152) and positively correlated with two apigenin derivates (Lmpp003930 and pme0368). There were 4 DEGs upregulated significantly in Y-R. *TGY013699* (*CsGSTF1*) was identified as a key candidate gene with a maximum FPKM ≥ 30. The expression level in Y-R was 3.1–7.7 folds higher than in other samples. It suggested that *CsGSTF1* might affect the yellow-red formation of leaves in tea plants by accelerating the transport process of flavonoid substrates.

The structural genes involved in the synthesis and transportation of flavonoids were regulated by transcription factors (TFs). The ternary complexes comprised by MYB-bHLH-WD40 were widely identified as essential regulators in the flavonoid pathway in plants [16]. Therefore, the relevant transcription factors of *MYB*, *bHLH*, and *WD40* were screened for further analysis. A total of 49 TFs, including 21 *MYB*, 23 *bHLH*, and 5 *WD40*, were obtained with the maximum FPKM ≥ 5. As shown in Figure 11a, the expression patterns of different members in each TF family varied widely. There was no clear regulatory mode in green and yellow leaves. Further correlation analysis revealed that 35 TFs were related to 13 DAMs of flavonoids (Figure 11b). Similarly, 18 genes were closely correlated with the three components (Hmmn004152, Lmpp003930, and pme0368). Moreover, the remaining 17 TFs were correlated with 8 DAMs of flavonoids. And the other two flavonoids (pmp000004 and Lmzp004885) showed the highest connectivity and were found to be correlated with 12 TFs. More importantly, *TGY012519 (MYB)* and *TGY000292 (bHLH)* were the key candidate genes due to their close relationship with the 5 flavonoids and connected the two correlation networks (marked in red font in Figure 11b), which indicated their potential roles in the color formation of tea leaves.

## 3. Discussion

### 3.1. There Are Differences in Primary and Secondary Metabolic Flux in Green and Yellow Leaves

The variations in leaf color are always accompanied by changes in metabolites. The metabolic fluxes were rearranged in varied leaf colors. Generally, weakened carbon metabolisms but enhanced nitrogen metabolisms were observed in yellow and albino tea leaves, including dramatically decreased chlorophyll and catechin contents and significantly increased amino acid contents, especially theanine [17]. The carbon metabolism in purple tea leaves was upregulated, which facilitates flavonoid/anthocyanin biosynthesis. Therefore, abundant bioactive compounds, particularly anthocyanins, were accumulated. These compounds not only have many benefits for human health but also play protective roles in responding to various biotic and abiotic stresses [18].

In this study, F1 half-sib populations of ‘Baijiguan’ with different colors were used as research materials. The chlorophyll accumulation in yellow leaves (Y-G, Y, and Y-R) was significantly lower than that in green leaves (Figure 1b). TEM analysis of the cellular ultrastructure further confirmed the defective chloroplast development with little or no stacked thylakoid structures formed in the yellow leaves (Figure 2e–h). The abnormal chloroplast structure directly destroyed the regular deposition of chlorophyll in cells and thus determined the leaf phenotype. This mechanism of leaf variation was widely confirmed in previous studies [19]. Interestingly, compared with the complete disruption of grana lamella in Y-G and Y, the granum structure stacked by thylakoid was partially reserved in Y-R. The corresponding SPAD value in Y-R increased accordingly, which was higher than that in Y-G and Y. It suggested the potential protective functions of red-related pigments in the maintenance of chloroplast structure [20]. Moreover, smaller vacuoles and more vesicles were also observed in yellow leaves, which is probably attributed to abnormal chloroplast development [5]. 

In general, different metabolites are assigned to specific organelles for synthesis and storage [21]. Chloroplast is the synthetic site of many primary metabolites, such as carbohydrates, amino acids, fatty acids, and vitamins. The synthesis of several secondary metabolites, such as hormones and pigments, is involved in the chloroplast envelopes [22]. Therefore, the impaired chloroplast is the main cause of the disordered carbon and nitrogen metabolism in albino tea plants [6]. 

Moreover, the defects in chloroplasts were always accompanied by the changed vacuole structure and appearance of many vesicular membrane structures [5,23]. Consistent results with smaller vacuoles and more vesicles were also observed in yellow leaves in this study (Figure 2a–d). The vacuole provides a storage site for many secondary metabolites. It plays an important role in regulating metabolic fluxes and maintaining metabolite homeostasis in cells [24]. Vesicles also participate in the transportation and compartmentation of secondary metabolites before accumulating in vacuoles [25]. 

Hence, the changes in cellular structure in variant colors of tea leaves definitely bring up a series of influences on metabolic pathways, especially for the distribution of pigment-related metabolites. It has been widely reported in diverse mutations of leaf colors in tea plants, including yellow (albino) or red (purple) germplasms [26,27,28]. The following metabolic profiling of varied colors in tea leaves re-confirmed the conclusion of our study. Both primary and secondary metabolites changed significantly along with the leaf color variations (Figure 3). It suggested that there are different directions of primary and secondary metabolic flux in green and yellow tea leaves [29]. 

Compared with the normal green leaf (G), a total of 47 DAMs divided into 11 classes were co-regulated significantly in Y-G, Y, and Y-R. Among them, 10 lipids were the dominant primary metabolites that were reduced obviously in yellow leaves, while 12 flavonoids were the representative secondary metabolites that were increased in yellow leaves (Figure 4). Consistent pathways were also co-enriched by further conjoint analysis of DAMs and DEGs (Figure 7a,b). It indicated that yellow leaves were inclined to have a weakened primary metabolism and an enhanced secondary metabolism [17]. Consistent variation trends were also verified in the normal green tea cultivar ‘Yinghong 9′ and its bud mutant ‘Huangyu’ with leaf etiolation [30]. The negative relationship between lipid and flavonoid pathways was implied [31]. It can be inferred that the repression in lipid metabolism changed the carbon flux, shifting to flavonoid metabolism in yellow leaves. It is notable that the flavonoid pathway also changed coordinately with the red color formation of tea leaves (Figure 4 and Figure 7c), which suggested its potential role in protecting chloroplast structure [32]. These results indicated that the color transformations of tea leaves among green, yellow, and red were closely linked with lipid and flavonoid metabolisms. 

### 3.2. Decrease in Lipid Metabolism Leads to the Disruption of Chloroplast Structure in Tea Leaves

Lipids are the essential structural constituents of cell membranes. Their biosynthetic process and homeostasis strongly affect chloroplast development. Thus, lipids play an important regulatory role in responding to various environmental changes [33]. Currently, the mechanisms of leaf color variation are widely investigated, but few studies have focused on the influences of lipid metabolism in the process of leaf color transformation [6]. Significant variation in lipid metabolism was found, particularly in ‘Baijiguan’, which was different from other albino tea mutants [12]. The leaf color of ‘Baijiguan’ under light treatment was regulated by light intensity by altering the contents and proportions of lipids. The proportion of unsaturated fatty acids in albino leaves was lower than that in shaded green leaves [11,12]. The changes in lipid metabolism and specific fatty acids affect the re-arrangement of the thylakoid membrane structure and the stability and functional maintenance of the membrane. It is an alternative strategy to reduce the impairment of cell membranes and enhance tolerance under environmental stresses, which has been demonstrated in many plants [34,35,36].

In our study, lipid-related pathways were consistently enriched by the conjoint analysis of the metabolome and transcriptome (Figure 7). Obviously, downregulated lipid metabolism in the yellow-green regions of variegated leaves than that in green regions was also found in *Pteroceltis tatarinowii* [37]. The relevant 10 DAMs were found to be decreased dramatically in etiolated leaves than the regular green leaves, especially for 4 LysoPE and 4 lipids (Figure 4). LysoPE belongs to glycerolipid, which constructs the bilayer membranes of thylakoids with various proteins, pigments, and photosynthetic cofactors embedded, and determines the chloroplast biogenesis [38,39]. The processes of fatty acid metabolism are also closely related to chloroplasts. They are synthesized in chloroplasts and then transferred to endoplasmic reticulum (ER) for triacylglycerol (TAG) production. Finally, they can be aggregated to form plastoglobuli in chloroplast [40]. In return, these fatty acids derived from chloroplast lipids function as important regulators in plant defense responses. The trafficking process of fatty acids from ER to chloroplast is necessary for the development and construction of chloroplast [41]. Therefore, it could be inferred that the decrease in lipid metabolism leads to the disruption of chloroplast structure in yellow tea leaves.

### 3.3. Increase in Flavones and Flavonols Metabolism Are Responsible for the Yellow Color Formation of Tea Leaves

The metabolism of flavonoids is associated with the leaf phenotype in tea plants [6,26]. Different components of flavonoids are involved in diverse color formations in plants. For example, anthocyanins are responsible for red or purple formation, while some flavones and flavonols are present as yellow pigments [42]. Previous studies on the albino tea leaves reported that flavonoid metabolism, mainly catechin biosynthesis, was highly inhibited [43]. However, few studies focus on other branched flavonoid pathways, such as flavones and flavonols. In our study, they were found to be enhanced in the yellow leaves. Consistent results were also observed in the yellow leaf mutant of ‘Rougui’ and ‘Anjibaicha’ [42,44]. Moreover, flavonols and flavones could function as antioxidative components. In an in vitro study of isolated chloroplasts, the oxidation of exogenous flavonol glycosides delayed the peroxidation of thylakoid lipids in illuminated chloroplasts [45]. It suggested that the flow of the branched flavonoid pathway was shifted to flavones and flavonols, which were responsible for the yellow color formation in tea leaves. This transitional direction in metabolic pathways also promoted the production of the antioxidant flavonoids, which could protect plants from ultraviolet damage in light stress and maintain the growth and development of mutants [20,27,46].

### 3.4. Modification, Transportation, and Transcription Factor Regulation of Flavones and Flavonols Also Affect Leaf Color in Different Yellow Tea Leaves

The increased concentrations of flavones and flavonols were found in yellow tea leaves, but most of the related genes involved in their biosynthetic pathway did not show consistent expression patterns. Therefore, there were other undetected genes that played an essential role in the flavonoid metabolism. 

The modification process of flavonoids is necessary for improving their solubility or stability and relieving the toxicity to cells. The glycosylation and acylation are generally modified formats in plants [47]. Most of the flavonoids, including anthocyanin, flavonols, and flavones were glucosylated by UDPG for stable storage of pigments in cells. Glucose, galactose, arabinose, and rhamnose are the main donors for flavonoid glycosylation [15]. HCT functions as an important acyltransferase to regulate the metabolic flux of phenylpropanoid and flavonoid pathways [48,49].

In our study, modification-related genes were upregulated in the etiolated leaves, especially in Y-G and Y. Two *UDPGs* (*TGY105792, TGY070785*) and one *HCT* (*TGY018791*) were identified for their potential roles in flavonoid glycosylation and acylation. *TGY105792* was annotated as flavonol 3-O-glucosyltransferase, a homologous gene to *UGT73B4* in Arabidopsis. It had significant catalytic activity toward flavonols-quercetin in vitro analysis [50]. *TGY105792* also showed similar sequences with *FaGT7* in *Fragaria×ananassa*, which was involved in the glucosylation of flavonols in fruits [51]. *TGY070785* encoded flavonol-3-O-glucoside/galactoside glucosyltransferase (*CsF3GGT*) and was found to have a high identity to *AcF3GGT1*, *GmF3G2”Gt*, and *In3GGT*. *AcF3GGT1* is responsible for the glycosylation of cyanidin. Thus, it affects the solubility and stability of the pigments and modifies the color of kiwifruit (*Actinidia chinensis*) [52]. *GmF3G2”Gt* had broad substrate activity for kaempferol/quercetin 3-O-glucoside/galactoside derivatives in soybean (*Glycine max*) [53]. *In3GGT* converts the anthocyanidin 3-O-glucosides into anthocyanidin 3-O-rutinosides by adding a rhamnose molecule. The allele gene with 4-bp insertion was found in all the yellow mutants of Japanese morning glory (*Ipomoea nil*) [54]. 

*TGY018791* was a member of the *HCT* gene family (*CsHCT*), and its protein sequence was homologous to AtHCT (Q9FI78.1) in Arabidopsis and NtHCT (Q8GSM7.1) in *Nicotiana tabacum*. They function as important acyltransferases to regulate the metabolic flux of the phenylpropanoid pathway—the upstream pathway of flavonoids [48,55]. In addition to catalyzing the reaction between p-coumaroyl-CoA and shikimate, they also showed broad substrate binding properties with phenolic compounds and managed their contents and compositions, such as lignin and chlorogenic acid [56,57]. It suggested that the modifying process of flavones and flavonols contributed to the related pigment accumulation and yellow formation in tea leaves. It may also be the repair strategy for reducing the negative effects of dysfunctional chloroplasts in plant cells [58,59]. 

Flavonoids are generally transported to the vacuoles for accumulation after modification. Two major efficient flavonoid transport systems, vesicle trafficking, and membrane transporters, were proposed in plants [15]. Transporter *TGY013699* was found to be significantly upregulated in Y-R. It was annotated as *TRANSPARENT TESTA 19* and encoded glutathione S-transferase (*CsGSTF1*). It showed sequence homology to *AtGSTF12.* Its transport functions of anthocyanin and flavan 3-ols for accumulation in vacuoles were verified in Arabidopsis [60,61]. The putative function of *AtGSTF12* in flavonol transportation was also proposed in the study of Kitamura [62]. *CsGSTF1* was proven to participate in the anthocyanin hyperaccumulation in a purple tea cultivar ‘Zijuan’ [28]. Moreover, the anomalously increased number of vesicles in Y-R also suggested a more active process of flavonoid transportation (Figure 2a–d) [61]. It suggested that *CsGSTF1* might affect the yellow-red formation of leaves in tea plants by accelerating the transport process of flavonoid substrate.

For TFs, *TGY012519* (*MYB*) and *TGY000292* (*bHLH*) were identified as key candidate genes involved in the regulation of flavonoid metabolism. *TGY000292* was annotated as *bHLH62* and significantly upregulated in Y-R. Two *bHLH62-like* genes were also found to differentially express in green, white, and purple pericarps in tea plants. It implied their potential roles in regulating color formation in tea plants [63]. *TGY012519* in Y-R showed 8.7–12.2 fold changes higher than other samples. It belonged to the R2R3 MYB transcription factor and was annotated as *anthocyanin 1* or *MYB1* (*CsAN1/CsMYB75*). It showed sequence homology with *AcMYB1* in kiwifruit (*Actinidia chinensis*). *AcMYB1* was responsible for the color changes in red-fleshed kiwifruit under temperature stress [64]. It also had effects on the accumulation of chlorophyll and carotenoids through transcriptional activation of metabolic pathway genes [65]. Moreover, coordinated regulation with relative transporters, transcription factors, and formation of vesicle-like bodies was proven to affect kiwifruit coloration [64,66]. *CsAN1/CsMYB75* was primarily considered an essential regulator of anthocyanin accumulation in purple tea leaves [67]. Further studies confirmed its interaction with the transporter CsGSTF1 and revealed a consistent molecular mechanism in regulating the color formation of tea leaves [28]. These results provided evidence for the integration function of transcription factors, *CsbHLH62* and *CsAN1/CsMYB75*, and transporter *CsGSTF1* in the red color formation of tea leaves.

## 4. Materials and Methods

### 4.1. Plant Materials

‘Baijiguan’ has a heritable etiolated phenotype in tender shoots under natural growing conditions. It was taken as a female parent, and its cutting seedlings were planted in the experimental tea garden of the Tea Research Institute of the Fujian Academy of Agricultural Sciences in Fu’an (119°34′39″ E, 27°12′57″ N), Fujian, China. The adjacent normal green tea varieties were taken as male parents. The hybrid seeds on ‘Baijiguan’ were obtained through open pollination and sown as individual strains. Then the F1 half-sibs of ‘Baijiguan’ were generated, which exhibited varied leaf colors under natural environments. In this study, four strains with distinctive leaf colors were used as materials (Figure 1a). Tea leaf samples with one bud and two leaves were collected on 7 April 2022, for metabolomic and transcriptomic profiling analysis. The first expanding leaf under the bud was used for chlorophyll determination and transmission electron microscopy analysis.

### 4.2. Determination of Leaf SPAD Value

The SPAD-502 chlorophyll meter was used for measuring leaf SPAD values (the relative chlorophyll content) in vivo. The average SPAD value of each leaf sample was performed for eight biological replicates. The significant difference was analyzed by one-way ANOVA with Duncan’s multiple range test (*p*-value < 0.05).

### 4.3. Transmission Electron Microscopy Analysis of Tea Leaves

Tender leaves were cut into thin strips after being plucked from tea plants and immediately fixed in a 2.5% glutaraldehyde solution for 2–4 h at room temperature, then stored at 4 °C for sample preparation. The fixed pieces were washed with 0.1 M phosphate buffer saline (PBS, pH 7.4) for 15 min and repeated two or three times. Then samples were re-fixed in 1% osmic acid containing 0.1 M PBS for 2 h and washed as mentioned above. The dehydrated samples were obtained through a gradient of ethanol solution (50–70–80–90–100–100%) for 15 min each time. Then the 1:1 mixture of acetone and 812 embedding agent was used for permeating samples overnight. Then samples were embedded for 48 h at 60 °C and cut into ultrathin slices (60–80 nm). After staining with a 2% uranium acetate saturated solution and lead citrate for 15 min, respectively, samples were finally subjected to ultrastructural observation using a JEM-1230 TEM (Tokyo, Japan).

### 4.4. Metabolomic Profiling Analysis

The relative metabolome and transcriptome analyses were accomplished by Metware Biotechnology Co., Ltd. (Wuhan, China). Details are as follows: 

Parts of the materials were dried by a vacuum dryer and ground into powders. 50 mg of samples were extracted with 1.2 mL 70% (*v*/*v*) methanol for 30 min and repeated 5 times. The final supernatant was acquired after centrifugation at 12,000 rpm for 3 min and filtered through 0.22 μm filters. 

The UPLC-ESI-MS/MS system was used for metabolite analysis. The chromatographic column was Agilent SB-C18 (1.8 μm, 2.1 mm × 100 mm). The mobile phase consisted of f solvent A, pure water with 0.1% formic acid, and solvent B, acetonitrile with 0.1% formic acid. The linear elution gradient was as follows: 0–9 min, 95–5% A; 10 min, 5% A; 11.1 min, 5–95% A; 14 min, 95% A. The column temperature was set to 40 °C. Samples with 4 μL injection volume were eluted at 0.35 mL/min flow rate. Each sample was performed for three independent extractions. 

PCA was performed by the statistics function prcomp within R (www.r-project.org, accessed on 13 October 2022). DAMs were screened by variable importance projection (VIP ≥ 1) and fold change (|Log2FC| ≥ 1.0).

### 4.5. Transcriptomic Profiling Analysis

Parts of the materials were fixed with liquid nitrogen immediately after being collected from plants. RNA extraction was performed using the RNAprep pure Plant Kit (Tiangen, Beijing, China). The quality and integrity of RNA was confirmed by a 1% agarose gel. The RNA concentration was determined by Biospec-Nano (Shimadzu, Kyoto, Japan). Then 1 μg RNA was used for library construction. The Illumina platform (Illumina, San Diego, CA, USA) was used for RNA sequencing. The raw data were filtered using fastp v 0.19.3. Then clean reads were subjected to the ‘Tieguanyin’ reference genome [68]. The gene expression level was calculated by FPKM. The DESeq2 method was used for difference analysis. Finally, DEGs were screened by setting a threshold with |Log2FC| ≥ 1.0 and false discovery rate (FDR) < 0.05.

To further verify the reliability of RNA sequencing data, 15 candidate DEGs were selected for qRT-PCR analysis. The primers were designed using the online software Primer-BLAST +2.13.0 (https://www.ncbi.nlm.nih.gov/tools/primer-blast/, accessed on 11 December 2022) and displayed in Appendix A. The Reverse transcription kit (R333) and SYBR qPCR master mix (Q712) from Nanjing Vazyme Biotech Co., Ltd. (Vazyme, Nanjing, China). were used for cDNA synthesis and qRT-PCR analysis, respectively. The experiment was conducted on the BIO-RAD Cycler CFX96 Real-Time PCR System (BIO-RAD, Hercules, CA, USA). GAPDH was used as the internal reference gene. The relative expression levels were calculated using the 2^−ΔΔCt^ method [69].

### 4.6. Conjoint Analysis of Metabolome and Transcriptome

In order to better understand the relationship between genes and metabolites, DEGs and DAMs in the same comparison group were mapped to the KEGG pathway. The top 25 (ranked by *p*-value) co-enriched pathways were displayed as a bubble map. Correlation analysis was performed using quantitative values of genes and metabolites in all samples. The Pearson correlation coefficient was calculated using the method ‘cor-function’ in R. The results were filtered by setting the threshold of correlation coefficient > 0.8 and *p*-value < 0.05. 

In addition, the heatmap of gene expression levels and correlation clustering were generated using the software TBtools v2.012 and Multi-Experiment Viewer, respectively. The correlation network of genes and metabolites was constructed by Cytoscape 3.5.1 software.

## 5. Conclusions

In summary, metabolic and transcriptomic differences among four closely related tea strains of ‘Baijiguan’ half-siblings with varied leaf colors were investigated. The directions of primary and secondary metabolic flux changed significantly along with the leaf color variations, and a putative regulatory mechanism was proposed. As shown in Figure 12, leaf color formation was influenced by the combined regulatory metabolism of lipids and flavonoid. Compared with G, the metabolic fluxes in etiolated leaves (Y-G, Y, and Y-R) were rearranged. They showed weaker lipid metabolism and enhanced flavonoid metabolism. On the one hand, the changes in lipid contents and proportions affect the re-arrangement of the thylakoid membrane structure. It leads to the disruption of chloroplast structure and decreased chlorophyll contents in etiolated leaves, which directly determine the formation of leaf colors. On the other hand, increased flavone and flavonol accumulation not only contributes to the leaf phenotype indirectly but also acts as an alternative protective mechanism in etiolated leaves. Moreover, different metabolic fluxes of the flavonoid pathway were also revealed in these etiolated leaves. Y-G and Y showed enhanced activities of the flavonoid modification process and increased content of eriodictyol and its derivates. While Y-R showed a vigorous process of flavonoid transportation and transcription factor regulation, increased contents of apigenin and its derivates. The key candidate genes, including UDPG, HCT, CsGSTF1, AN1/CsMYB75, and bHLH62, might play important roles in this process. This study provides new insights into the variation mechanisms of different leaf colors in tea plants.

## Figures and Tables

**Figure 1 ijms-25-00242-f001:**
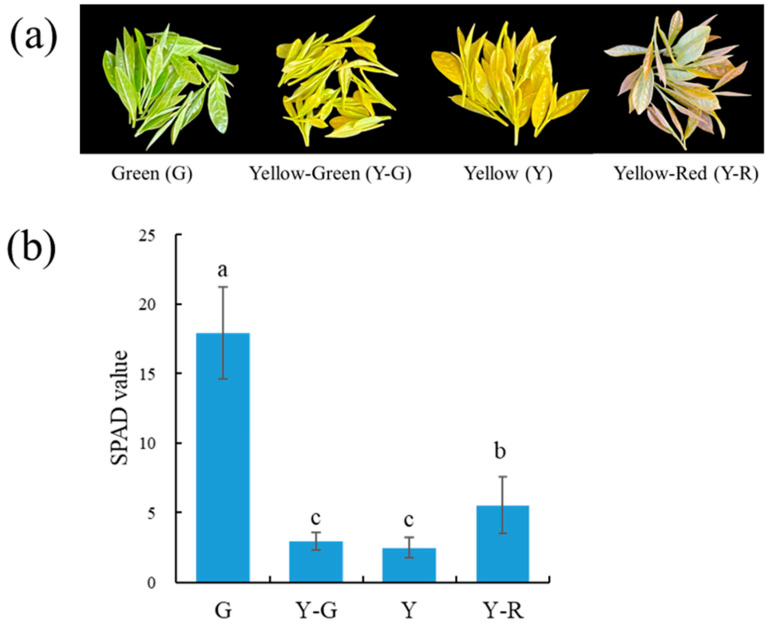
The phenotypes (**a**) and SPAD values (**b**) of tea leaves with varied colors. Data are mean ± SD from eight biological replicates. The different letters indicated a significant difference at *p* < 0.05.

**Figure 2 ijms-25-00242-f002:**
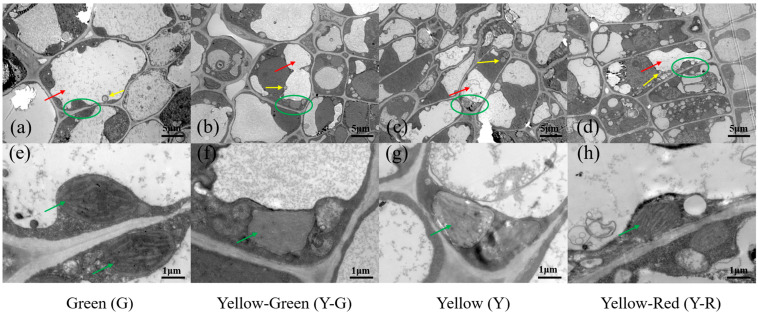
The cellular ultrastructure of tea leaves in green leaves (**a**,**e**), yellow-green leaves (**b**,**f**), yellow leaves (**c**,**g**) and yellow-red leaves (**d**,**h**). Chloroplasts were marked in green circles in (**a**–**d**) and green arrows in (**e**–**h**); Vacuoles were marked in red arrows in (**a**–**d**); Vesicles were marked in yellow arrows in (**a**–**d**).

**Figure 3 ijms-25-00242-f003:**
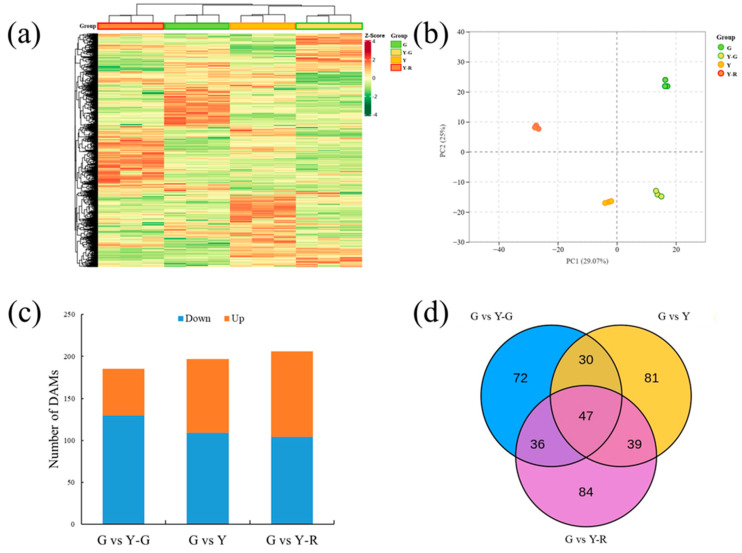
Metabolomic profiling analysis of tea leaves with different colors. (**a**) Heatmap of all metabolites detected; (**b**) Principal component analysis of metabolomic data; (**c**) Differentially accumulated metabolites analysis using the G group as control. (**d**) Venn analysis of differentially accumulated metabolites.

**Figure 4 ijms-25-00242-f004:**
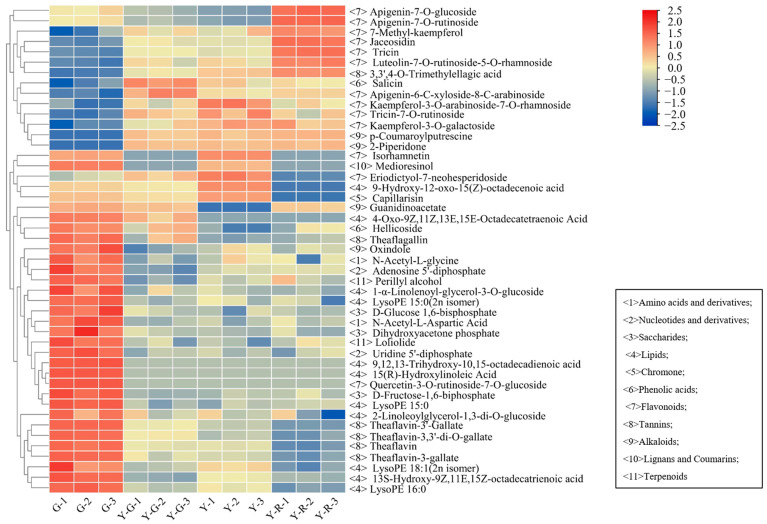
The distributed patterns of 47 common differentially accumulated metabolites.

**Figure 5 ijms-25-00242-f005:**
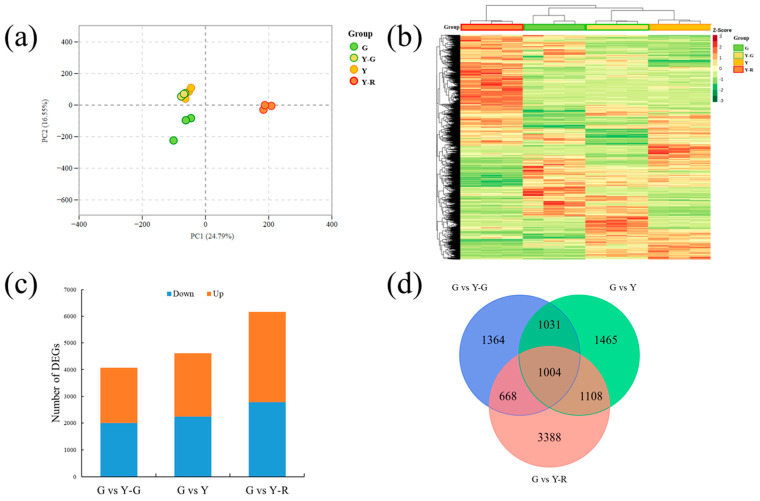
Transcriptomic profiling analysis of tea leaves with different colors. (**a**) Principal component analysis of transcriptomic data; (**b**) Heatmap of different expression genes; (**c**) The number of different expression genes in three comparisons. (**d**) Venn analysis of different expression genes.

**Figure 6 ijms-25-00242-f006:**
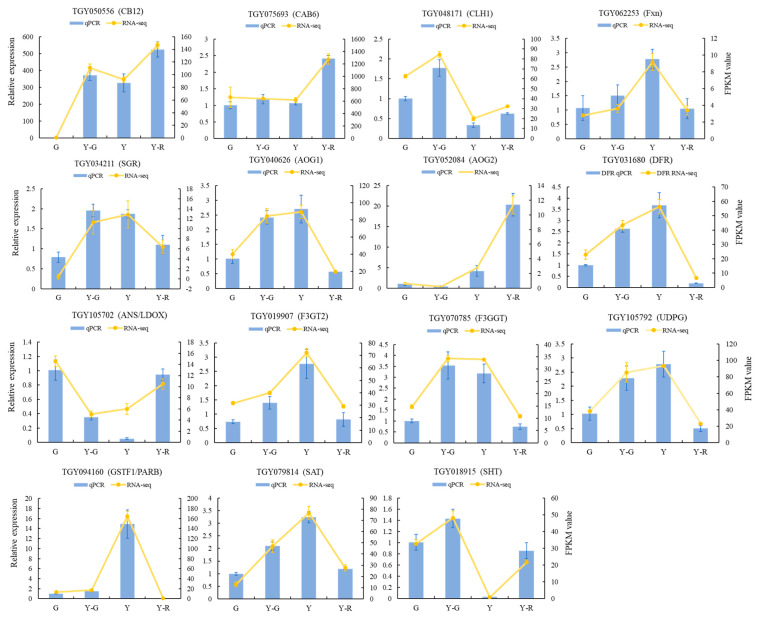
Validation of transcriptomic data by qRT-PCR analysis. Data are mean ± SD from three biological replicates.

**Figure 7 ijms-25-00242-f007:**
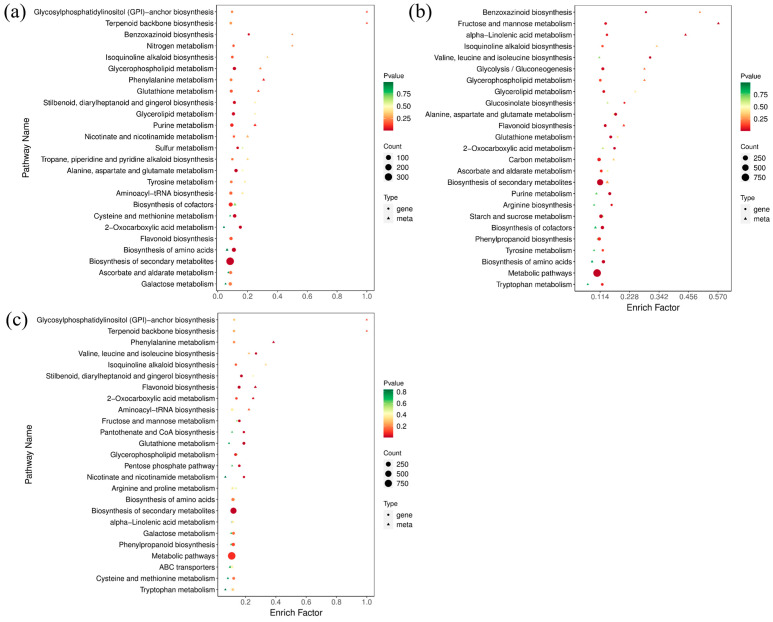
The top 25 co-enriched KEGG pathways in the comparison of G vs. Y (**a**), G vs. Y-R (**b**), and Y vs. Y-R (**c**).

**Figure 8 ijms-25-00242-f008:**
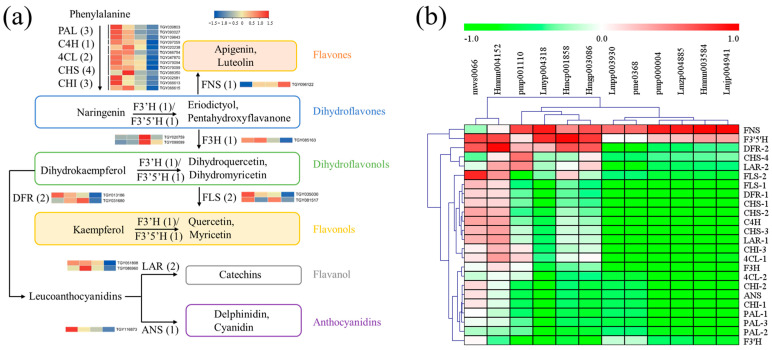
Functional biosynthetic genes are involved in the flavonoid pathway. (**a**) Heatmap of these gene expression patterns; (**b**) Correlation analysis between gene expression and related flavonoid contents. PAL, phenylalanine ammonia-lyase; C4H, cinnamate-4-hydroxylase; 4CL, 4-coumaroy CoA ligase; CHS, chalcone synthase; CHI, chalcone isomerase; F3′H, flavonoid 3′ hydroxylase; F3′5′H, flavonoid 3′,5′-hydroxylase; F3H, flavanone-3-hydroxylase; FNS, flavone synthase II; FLS, flavonol synthase; DFR, dihydroflavonol 4-reductase; ANS, anthocyanidin synthase; LAR, leucoanthocyanidin reductase.

**Figure 9 ijms-25-00242-f009:**
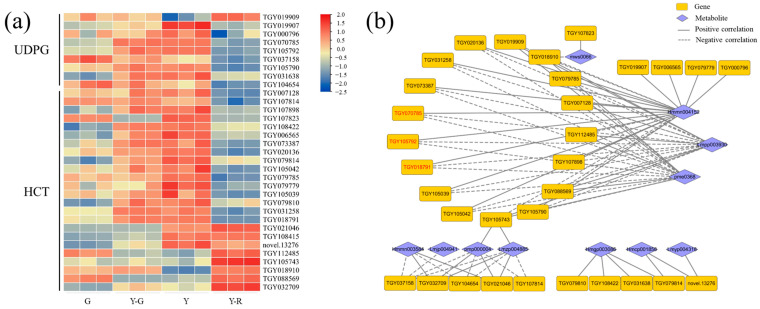
Analysis of flavonoid modification-related genes. (**a**) The expression patterns of glycosylated and acylated genes; (**b**) Correlation network of flavonoid modification-related genes and metabolites. UDPG, UDP-glycosyltransferase; HCT, hydroxy-cinnamoyl transferase.

**Figure 10 ijms-25-00242-f010:**
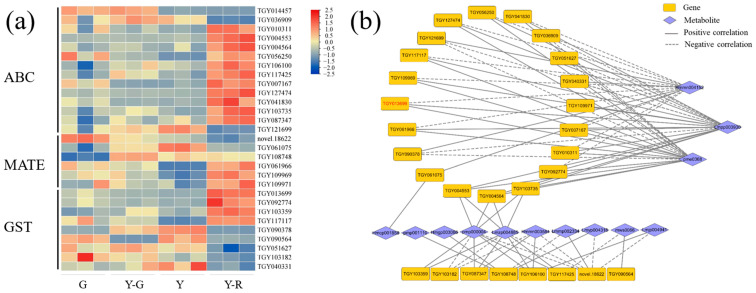
Analysis of potential flavonoid transporters. (**a**) The expression patterns of ABC, MATE, and GST transporters; (**b**) Correlation network of flavonoid transportation-related genes and metabolites. ABC, ATP-binding cassette; MATE, multidrug and toxin extrusion; GST, glutathione S-transferase.

**Figure 11 ijms-25-00242-f011:**
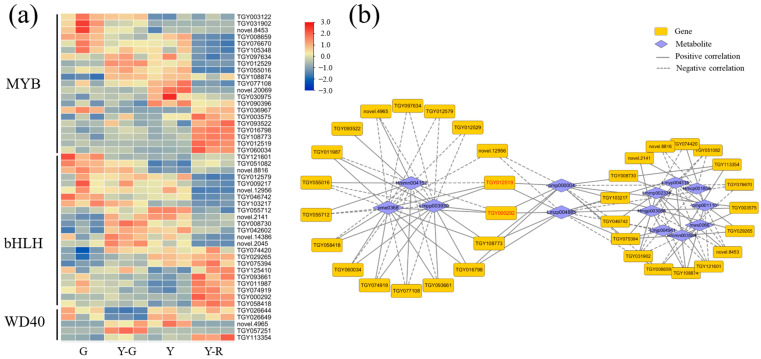
Analysis of potential transcription factors involved in the flavonoid pathway. (**a**) The expression patterns of MYB, bHLH, and WD40; (**b**) Correlation network of transcription factors and related flavonoids.

**Figure 12 ijms-25-00242-f012:**
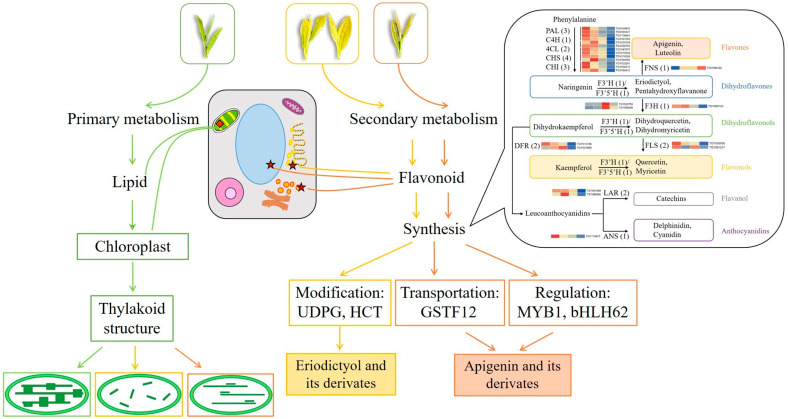
Proposed mechanisms of leaf color variations in the offspring of ‘Baijiguan’.

## Data Availability

The dataset generated during and/or analyzed during the current study is available from the corresponding author on reasonable request.

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
