# Peer review of "Integrated Analysis of Metabolome and Transcriptome Revealed Different Regulatory Networks of Metabolic Flux in Tea Plants [Camellia sinensis (L.) O. Kuntze] with Varied Leaf Colors"

_ijms, 2023, doi:10.3390/ijms25010242_

Round 1

Reviewer 1 Report

Comments and Suggestions for Authors

This paper characterizes leaf coloration in four strains of the "Baijiguan" F1 half-sib generation using a metabolomics and transcriptomic approach. I believe that the results presented in this article will be valuable for the tea research community. However, I feel that the current manuscript does not meet the standards required for publication as a scientific research paper. One of the major issues is the lack of information on materials and experimental conditions. Without such information, it will be impossible for others to replicate the results reported in this paper. I kindly request the authors to revise the manuscript to provide detailed descriptions of the materials and methods.

1 Plant materials

The authors referred to the plant materials used as "closely related tea strains" of the tea cultivar Baijiguan (line 70). However, I could not find any detailed information, including the genetic background of these strains analyzed in this study. It is problematic to refer to the materials simply as green (G), yellow-green (Y-G), yellow (Y), and yellow-red (Y-R). Do these strains have specific cultivar names or strain numbers? How were they obtained from Baijiguan? How closely related are they genetically to each other? I strongly recommend that the authors provide more detailed information about the four strains of tea leaves used in this study.

2 Sampling conditions

The authors mentioned light-sensitive mutants of Baijiguan, but there is no description of light intensity in the materials and methods. The lack of such information makes it difficult to interpret the electron microscopy (EM) photographs as shown in Figure 3. Depending on the duration of shading, they might have regreened under low light intensity. This aspect is also critical for interpreting the results of changes in secondary metabolites and gene expression since some of them are constitutive or housekeeping, while others are transient or stress-responsive. Therefore, I kindly request that the authors provide more details on the sample processing procedures and conditions.

3 Proposed mechanism

The proposed mechanisms (Figure 12) do not contribute to understanding the mechanisms behind color variations. I suggest removing this premature presentation. Considering its light sensitivity, oxidative stress due to photoinhibition of photosynthesis could be involved in the primary mechanism, as reported previously. In this context, ascorbate deficiency should be the primary cause of the induction of secondary metabolites, including flavonoid biosynthesis (the flavonoid antioxidant hypothesis). While ascorbate content may not be significant in the case of oolong tea, it strongly affects the taste and flavor of green tea.

Author Response

1 Plant materials

The authors referred to the plant materials used as "closely related tea strains" of the tea cultivar Baijiguan (line 70). However, I could not find any detailed information, including the genetic background of these strains analyzed in this study. It is problematic to refer to the materials simply as green (G), yellow-green (Y-G), yellow (Y), and yellow-red (Y-R). Do these strains have specific cultivar names or strain numbers? How were they obtained from Baijiguan? How closely related are they genetically to each other? I strongly recommend that the authors provide more detailed information about the four strains of tea leaves used in this study.

Response 1: Good suggestions. ‘Baijiguan’ has heritable etiolated phenotype in tender shoots under natural growing conditions, which acted as a female parent in our study. The adjacent normal green tea varieties were taken as male parents. The hybrid seeds on ‘Baijiguan’ were obtained through open pollination and sown as individual strain. Then the F1 half-sibs of ‘Baijiguan’ were generated, which exhibited varied leaf colors under an identical natural environment. In this study, four strains with distinctive leaf colors were used as materials (Fig 1a). Detailed information about plant materials were provided (line 81-84, line 509-520).

2 Sampling conditions

The authors mentioned light-sensitive mutants of Baijiguan, but there is no description of light intensity in the materials and methods. The lack of such information makes it difficult to interpret the electron microscopy (EM) photographs as shown in Figure 3. Depending on the duration of shading, they might have regreened under low light intensity. This aspect is also critical for interpreting the results of changes in secondary metabolites and gene expression since some of them are constitutive or housekeeping, while others are transient or stress-responsive. Therefore, I kindly request that the authors provide more details on the sample processing procedures and conditions.

Response 2: You are right and more details on the sample processing procedures and conditions were provided (line 81-84, line 509-520). The etiolated phenotype of tender leaves in ‘Baijiguan’ was observed under natural growing conditions. Its F1 half-sib offsprings also exhibited varied leaf colors under identical natural environmental condition. So, materials used in this study were collected directly without any other treatments.

3 Proposed mechanism

The proposed mechanisms (Figure 12) do not contribute to understanding the mechanisms behind color variations. I suggest removing this premature presentation. Considering its light sensitivity, oxidative stress due to photoinhibition of photosynthesis could be involved in the primary mechanism, as reported previously. In this context, ascorbate deficiency should be the primary cause of the induction of secondary metabolites, including flavonoid biosynthesis (the flavonoid antioxidant hypothesis). While ascorbate content may not be significant in the case of oolong tea, it strongly affects the taste and flavor of green tea.

Response 3: Many thanks. In this study, fresh tea leaves with different colors were used as materials without additional light treatments or any tea processing procedures. Sorry about the misunderstanding for lack of information on materials and experimental conditions.

The proposed mechanisms in Figure 12 showed that the leaf color formation was influenced by the combined regulatory metabolism of lipid and flavonoid. More detailed descriptions were provided in the part of ‘5. Conclusions’ at line 590-610. Relevant revisions were also made in the part of ‘Abstract’ at line 13-29.

Reviewer 2 Report

Comments and Suggestions for Authors

The complex metabolic pathways of flavonoid, terpenoid, and lipid synthesis play crucial roles in plant biology, influencing various physiological processes and contributing to the final color features of plants. In the phenylpropanoid pathway, enzymes such as phenylalanine ammonia-lyase (PAL), chalcone synthase (CHS), chalcone isomerase (CHI), and flavanone 3-hydroxylase (F3H) among many others are crucial in catalyzing specific reactions leading to the accumulation of flavonoids, such as anthocyanins (red, purple, or blue pigments), flavonols, and flavones, largely contribute the final color of plant tissues. Moreover, transcription factors like MYB and bHLH play a crucial role in the transcriptional regulation of many genes involved in flavonoid biosynthesis, which was confirmed in the present paper. The synthesis of terpenoids is also involved in color determination, with final metabolites like carotenoids (e.g. beta-carotene and lycopene) contributing to the yellow, orange, and red colors in plants. The presence of lipids, especially those associated with chloroplast membranes, can influence leaf color due to their interaction with pigments and thylakoid structure. Lipid metabolism is tightly regulated by key enzymes like acetyl-CoA carboxylase (ACC) and fatty acid synthase (FAS), but also by environmental factors (including light), developmental stages, and plant hormones.

Because the pathways of biosynthesis of flavonoids, terpenoids, and lipids are interconnected, their metabolic fluxes can influence each other. Lipids, particularly those associated with the thylakoid membrane, play a crucial role in the stabilization and interaction with chlorophyll and other pigments, influencing the overall coloration. Light conditions, temperature, and nutrient availability are external factors that can modulate the activities of these metabolic pathways, thereby affecting plant color. In summary, the final color features of plants are a result of the intricate interplay of these metabolic pathways, their regulation, and the environmental conditions.

The article is original and can contribute to the general knowledge about the mechanisms that underlay the variation in color pigment (chlorophyll) in tea plants and can be used to select lines that share specific leaf color features. The secondary metabolites are of great importance also for plant development and protection against harmful pathogens of pests. The most interesting finding of this research is the fact, that the reduced content of unsaturated fatty acids contributed to the “clearer” color of leaves besides the reduced green (chlorophyll) pigment in the Y variant (which is not surprising itself). The experimental design is appropriate, the methodology used in transcriptomic profiling analysis and the metabolome and transcriptome processing do not raise any concern. The final Figure no 12 well resumes the mechanism of leaf color variation in the tea plant examined.

I have no particular comments because the general scientific approach is properly stated and well explained. Besides the molecular aspect of the presence or absence of some biochemical compounds, the study focused also on cell inside structures, i.e. presence of vesicles, the shape of vacuoles, distribution of pigment-related metabolite, which gives a complete image of the narration. The discussion covered the complex metabolic pathways of flavonoid, terpenoid, and lipid synthesis, with emphasis on the crucial points that can be involved in the final color feature as a result. References are complete, and most of them were published during the last 20 years.

Nevertheless, two improvements can be made:

The visualization of figures, in lines 182 Figure 6, and 196 Figure 7– can be improved with better quality and more readable descriptions (which are too small).

In Materials and Methods. Plant material could be better described, with geographical coordinates of the collection site, and giving ambient light conditions (i.e. photoperiod duration) because it can play a role in carotenoid compound accumulation in the etiolated plant, a light-sensitive albino ‘Baijiguan’ cultivar.

I recommend publishing the article after that correction.

Author Response

Two improvements can be made:

The visualization of figures, in lines 182 Figure 6, and 196 Figure 7– can be improved with better quality and more readable descriptions (which are too small).

Response 1: Good suggestions. The Figure 6 and Figure 7 were improved and enlarged (line 203 and line 219).

In Materials and Methods. Plant material could be better described, with geographical coordinates of the collection site, and giving ambient light conditions (i.e. photoperiod duration) because it can play a role in carotenoid compound accumulation in the etiolated plant, a light-sensitive albino ‘Baijiguan’ cultivar.

Response 2: You are right. More detailed description on plant materials were provided in the part of ‘4.1 Plant materials’ (line509-520)

Reviewer 3 Report

Comments and Suggestions for Authors

Comments for the manuscript entitled "Integrated analysis of metabolome and transcriptome revealed different regulatory networks of metabolic flux in tea plants (Camellia sinensis) with varied leaf colors" submitted by Yazhen Zhang et al.

This study is interesting because it extensively investigates the molecular mechanism of color formation of tea leaves. The research started from four strains from 'Baijiguan' (tea cultivar - an impotant parent for genetic breeding) F1 half-sib generation with different colors: Green (G), Yellow-Green (Y-G), Yellow (Y), Yellow-Red (Y-R). The investigations included the cellular ultrastructure of tea leaves, highlighting the abnormal morphology of cells and chloroplasts at the leaves of Y-G, Y, Y-R. 1146 metabolites were detected using the metabolome technology, 47 of which were common to the four phenotypes, but had differential accumulation for Y-G, Y and Y-R compared to G. Lipids ( primary metabolites) had the highest levels in G, and flavonoids (secondary metabolites) recorded high levels in Y-G, Y Y-R. Both primary and secondary metabolites changed significantly with leaf color variations. Between the lipid path and the flavonoid path, a negative relationship was highlighted.

For the discovery of the molecular mechanism of color formation in tea leaves, the authors applied RNA-seq analysis, followed by real-time PCR analysis (QRT-PCR). The analysis of the metabolome and transcriptome allowed the discovery of the mechanism of regulation of the color of the tea leaves. In the leaves Y-G, Y and Y-R, higher levels of gene expression were found than in G. The authors indicate 4 candidate genes (UDPG, HCT, CsGSTF1, AN1/CsMYB75, bHLH62) that would have key roles in the path of flavonoids. The structural genes involved in the synthesis and transport of flavonoids were regulated by transcription factors that were examined for additional analysis. The authors propose molecular mechanisms that explain the color variations of tea leaves, based on the combined metabolism of lipid and flavonoid regulation, pointing to the fact that the accumulation of flavone and  flavonols in Y-G, Y, Y-R can be a protective mechanism.

These researches provide insights into the mechanism of tea leaf color variation.

My comments are below:

1. The correct scientific name for Camellia sinensis is: Camellia sinensis (L.) Kuntze. Therefore, add to the title of the paper.

2. The abbreviation SPAD should be explained for the public less familiar with chlorophyll content determination. You only did this in Materials and Methods, but you should also do it when you use it for the first time.

3. In line 395, "In the vitro study" is not correct. Should: In vitro study.

4. In line 474, you wrote: "They were planted ...". Should: "They were sown". The seeds are sown, and the cuttings, potato tubers, trees, and so on are planted!

5. In line 480, you should write in vivo with italics.

6. It would be better to move Figure 12 to the end of the "Results" section. In this sense, I suggest you create subsection 2.7. Proposed  mechanisms of leaf color variations in the offsprings of 'Baijiguan'. Make a short comment on the proposed mechanism, then add Figure 12.

7. Additional Tables (S1,  S2, S3, S4) can be placed directly in the article.

I wish you success in publishing this study!

Author Response

1. The correct scientific name for Camellia sinensis is: Camellia sinensis (L.) Kuntze. Therefore, add to the title of the paper.

Response 1: You are right. ‘Camellia sinensis (L.) O. Kuntze’ was added to the title of the paper.

2. The abbreviation SPAD should be explained for the public less familiar with chlorophyll content determination. You only did this in Materials and Methods, but you should also do it when you use it for the first time.

Response 2: Good suggestions. The abbreviation ‘SPAD’ was explained and revised in line 89-94.

3. In line 395, "In the vitro study" is not correct. Should: In vitro study.

Response 3: Thanks. It was revised in line 420.

4. In line 474, you wrote: "They were planted ...". Should: "They were sown". The seeds are sown, and the cuttings, potato tubers, trees, and so on are planted!

Response 4: You are right. It was revised in line 509-520.

5. In line 480, you should write in vivo with italics.

Response 5: Thanks. It was revised in line 523.

6. It would be better to move Figure 12 to the end of the "Results" section. In this sense, I suggest you create subsection 2.7. Proposed mechanisms of leaf color variations in the offsprings of 'Baijiguan'. Make a short comment on the proposed mechanism, then add Figure 12.

Response 6: Many thanks. Based on the previous studies and further analysis in the ‘3. Discussion’, the comprehensive regulatory network of metabolic flux in varied leaf colors was revealed more clearly. Then the mechanism was proposed and displayed as Figure 12. Therefore, it was more suitable to put Figure 12 after the ‘3. Discussion’ section. Finally, it was put in the ‘5. Conclusion’ section.

7. Additional Tables (S1, S2, S3, S4) can be placed directly in the article.

Response 7: Thanks. Because the additional tables were too big to be placed directly in the article, they were re-submitted as excel files to the website.

Round 2

Reviewer 1 Report

Comments and Suggestions for Authors

After incorporating the comments and suggestions of the reviewers, the authors have revised the manuscript, resulting in significant improvements. Although the manuscript still lacks information on light conditions, as pointed out by two reviewers, the overall revision is satisfactory.